# Cloze Evaluation for Deeper Understanding of Commonsense Stories in Indonesian

**Fajri Koto**[1]     **Timothy Baldwin**[1,2]     **Jey Han Lau**[1]

[1]The University of Melbourne
[2]MBZUAI

`ffajri@student.unimelb.edu.au, tb@ldwin.net, jeyhan.lau@gmail.com`

## Abstract

Story comprehension that involves complex causal and temporal relations is a critical task in NLP, but previous studies have focused predominantly on English, leaving open the question of how the findings generalize to other languages, such as Indonesian. In this paper, we follow the Story Cloze Test framework of Mostafazadeh et al. (2016) in evaluating story understanding in Indonesian, by constructing a four-sentence story with one correct ending and one incorrect ending. To investigate commonsense knowledge acquisition in language models, we experimented with: (1) a classification task to predict the correct ending; and (2) a generation task to complete the story with a single sentence. We investigate these tasks in two settings: (i) monolingual training and (ii) zero-shot cross-lingual transfer between Indonesian and English.

## 1 Introduction

Commonsense reasoning is a key component of natural language understanding (NLU), which previous work (Charniak, 1972; Mueller, 2004; Mostafazadeh et al., 2016; Chen et al., 2019) has attempted to model through tasks such as story comprehension. While humans can easily comprehend temporal and causal relations to understand a story narrative, machines tend to struggle due to implicit information and story premises. Often, *world knowledge* such as social conventions, the laws of nature, and common logic are required to connect the premises to draw appropriate conclusions or closure (Shoham, 1990; Ponti et al., 2020).

Mostafazadeh et al. (2016) and Sharma et al. (2018) introduced the *Story Cloze Test* framework to empirically evaluate commonsense reasoning, based on English short stories about daily-life events. The task is to choose the correct ending of a four-sentence story based on a two-way multiple choice. Mostafazadeh et al. (2016) published 3,700 data pairs, and the dataset has been used to model

commonsense reasoning (Schwartz et al., 2017; Liu et al., 2018; Sap et al., 2019; Chen et al., 2019; Li et al., 2019) and perform discourse probing of pretrained language models (Koto et al., 2021).

There is a lack of research modeling story comprehension in languages beyond English. Ponti et al. (2020) argued that current progress over English may not generalize to other languages because of its Anglocentric bias both linguistically, and also in terms of cultural and social conventions (Thomas, 1983). Motivated by this, we explore commonsense reasoning in Indonesian by constructing a dataset based on the framework of Mostafazadeh et al. (2016).

XCOPA (Ponti et al., 2020) is perhaps the most closely-related work to ours, wherein 600 instances of the COPA dataset (Roemmele et al., 2011) were manually translated into 11 languages, including Indonesian. COPA is an open-domain commonsense causal reasoning task that consists of two-sentence pairs, and does not include complex narrative comprehension. Moreover, the translation approach also has its own limitations, in entrenching Anglocentric social contexts in other languages.

To summarize, we introduce the first Story Cloze Test in Indonesian, and perform preliminary studies based on: (1) a classification task to predict the correct ending (Li et al., 2019); and (2) a single-sentence generation task to complete the story (Guan et al., 2019; Huang et al., 2021). We perform these two tasks in two settings: (1) monolingual training, and (2) zero-shot cross-lingual transfer, between Indonesian and English. Our data and code are available at https://github.com/fajri91/IndoCloze.

## 2 Dataset Construction

Following Mostafazadeh et al. (2016), we construct an Indonesian Story Cloze Test dataset. Each instance consists of a four-sentence premise, and two candidates for the fifth sentence: an appropriate

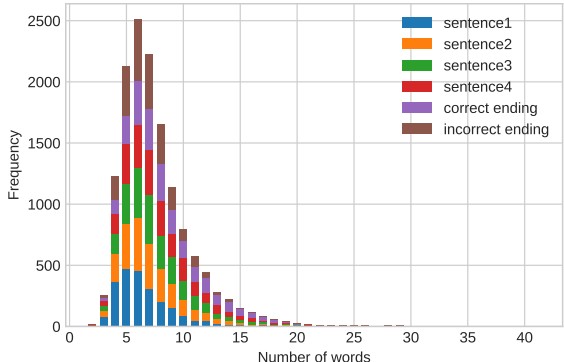

Figure 1: Number of words in each sentence position.

| Person (#unique: 1962) | Location (#unique: 114) | Organization (#unique: 166) |
|---|---|---|
| Rio, Acha, Reno, Mamat, Hana, Gina, Juju, Tarra, Maria, Elisa | Indonesia, Jakarta, Bandung, Kenya, Bali, Jogja, Surabaya, Korea, Monas | SD Harapan, KAI, SMA Harapan, SMA Angkasa, Bobo, Bimbel, SMP Harapan |

Table 1: Examples of PERSON, LOCATION, and ORGANIZATION (sampled from top-20 predictions).

and inappropriate ending. Similar to Mostafazadeh et al. (2016) and Sharma et al. (2018), our corpus consists of daily-life events, but in Indonesian contexts (e.g. locations, places, names, food, culture).

**Data creation.** We hired seven Indonesian university students to each write 500 short stories over a period of one month. As part of the recruitment, candidates were provided with story requirements and several examples,[1] and asked to write a 5-sentence story, as well as an inappropriate fifth sentence. From ten applicants, we hired the seven best candidates based on their submitted stories. After one month, four workers completed the job and were paid Rp 750,000.[2] The three who did not complete the task were paid a prorated salary, based on the number of completed stories. This resulted in a dataset of 2,335 stories (see Table 2 for examples).

**Quality control.** We additionally assessed the dataset by employing two Indonesian university students that were not involved in the data construction.[3] Based on 100 random samples, we asked each worker to choose the correct fifth sentence for a given four-sentence premise, and found that both

workers achieved 99% accuracy.[4]

**Data statistics.** Our corpus contains 14,010 sentences and 106,479 words. In Figure 1, we observe that word counts in each sentence position are somewhat similar, with a median sentence length of 5–10 words.

We used an IndoBERT model (Koto et al., 2020) to train POS and NER models, based on the datasets of Dinakaramani et al. (2014) and Gultom and Wibowo (2017), resp., and used them to predict VERB, PERSON, LOCATION, and ORGANIZATION tags.[5] First, we found that the dataset contains 21,447 VERB tokens (3,723 unique tokens), with the top-3 most frequent verbs having a frequency of 2% (see Figure 2 in Appendix). We also observe that PERSON, LOCATION, and ORGANIZATION NEs are mostly local Indonesian expressions, with common PERSON names being *Reno* and *Mamat*, and organization names being *KAI* and *Bobo*, as captured in Table 1. Additionally, we found that the top-5 most frequent bigrams and trigrams have a frequency of less than 0.3%, demonstrating the lexical diversity of our stories, even though the dataset was created by a small number of workers (Table 3).

## 3 Experimental Setup

Similar to Bhagavatula et al. (2020) experiments in English commonsense reasoning, we conducted two tasks: (1) a classification task to predict the correct ending; and (2) a single-sentence generation task to complete the story. We perform these two tasks in two settings: (1) monolingual training, and (2) zero-shot cross-lingual transfer, between Indonesian and English. The data split is presented in Table 4.

### 3.1 Classification

Following Mostafazadeh et al. (2016), we evaluate the classification task based on accuracy, defined as $\frac{\#correct}{\#testcases}$. Models are tuned based on the development set, and results are averaged over three runs. We experiment with the following four models.

$n$-**gram overlap**: We select candidate with the highest ROUGE-1 (F1; Lin (2004)), computed between the premise and ending.

**fastText-based similarity**: We pick the candidate with the highest cosine similarity, computed

---

[1] See Appendix for more details.

[2] The monthly minimum wage in Indonesia is around Rp 4,000,000, and the workload to write 500 short stories equates to roughly 5-days of full-time work.

[3] We paid Rp 150,000 to each.

[4] The two candidate fifth sentences (the correct and incorrect endings) are shuffled for each story.

[5] The POS and NER models have accuracies of 96.8% and 90.1%, respectively.

| | Indonesian | English |
|---|---|---|
| **Context** | *Sepulang sekolah, Rani dan Rina mengunjungi toko komik. Komik kesukaan mereka terbit hari ini. Masing-masing membayar sepuluh ribu rupiah. Setelah membayar, mereka berdua pulang ke rumah* | After school, Rani and Rina visit a comic shop. Their favorite comic will be published today. Each of them paid ten thousand rupiah. After paying, the two of them went home. |
| **Right ending** | *Mereka membaca komik itu bersama-sama di rumah.* | They read the comic together at home. |
| **Wrong ending** | *Komik itu mereka robek jadi dua bagian.* | They tore the comic into two parts. |
| **Context** | *Hari ini langit sangat mendung. Gemuruh sudah terdengar sejak pagi. Diprediksi hujan akan segera turun. Aku bergegas berangkat kerja karena takut kehujanan.* | Today the sky is very cloudy. There has been thunder since morning. It is predicted that rain will fall soon. I rush to work to avoid the rain. |
| **Right ending** | *Aku membawa jas hujan.* | I take a raincoat. |
| **Wrong ending** | *Sebelum berangkat, aku menjemur pakaian di halaman rumah* | Before leaving, I hang my washing outdoors. |
| **Context** | *Boni punya 5 balon. Balon ini dibelikan oleh ayah di Jalan Margonda. Semua balon Boni berbeda. 2 balon berwarna merah dan biru.* | Boni has 5 balloons. These balloons were bought by his father at Jalan Margonda. All Boni's balloons are different colours. Two of the balloons are red and blue. |
| **Right ending** | *Yang lain berwarna putih, hitam, dan kuning* | The others are white, black and yellow. |
| **Wrong ending** | *Sedangkan ketiga lainnya berwarna merah muda.* | While the other three are pink. |

Table 2: Three example Story Cloze Test instances, with an English translation for illustrative purposes.

| Bigram (#unique: 59,256) | Freq (%) |
|---|---|
| *pergi ke* (go to) | 0.30 |
| *tidak bisa* (can not) | 0.29 |
| *hari ini* (today) | 0.27 |
| *teman temannya* (his/her friends) | 0.25 |
| *tidak pernah* (never) | 0.25 |
| **Trigram (#unique: 72,443)** | **Freq (%)** |
| *oleh karena itu* (therefore/thus) | 0.04 |
| *pulang ke rumah* (go home) | 0.04 |
| *dengan teman temannya* (with his/her friends) | 0.03 |
| *maka dari itu* (therefore/thus) | 0.03 |
| *dan teman temannya* (and his/her friends) | 0.03 |

Table 3: Top-5 bigrams and trigrams.

| Task | EN | ID (ours) |
|---|---|---|
| Classification | 1,683 / 188 / 1,871 | 1,000 / 200 / 1,135 |
| Generation | 45,496 / 1,871 / 1,871 | 1,000 / 200 / 1,135 |

Table 4: Data distribution of train/development/test set. The English dataset is from Mostafazadeh et al. (2016).

between the premise and ending based on 300d Indonesian `fastText` (Bojanowski et al., 2017).

**Hierarchical BiLSTM**: We use a two-level 200d BiLSTM, using the first to encode a single sentence with 300d `fastText` as input. We perform average pooling to obtain a sentence representation, and apply the second BiLSTM across all sentences. We concatenate the last hidden state of the two LSTMs, and perform binary classification using a sigmoid function (see Appendix for hyper-parameters).

**Pretrained Language Models**: We fine-tune MBERT (Devlin et al., 2019) and INDOBERT

(Koto et al., 2020) by concatenating the premise and ending sentence, and use `[CLS]` for classification (see Appendix for hyper-parameters).[6]

For classification, we first evaluate the difficulty of our dataset by predicting the fifth sentence based on a different combination of premises as context. For zero-shot cross-lingual transfer, we use the English corpus of Mostafazadeh et al. (2016), and also use translations from Google Translate.[7]

### 3.2 Generation

We use the four-sentence premise as input, and train MBART (Liu et al., 2020) to generate the fifth sentence for both English and Indonesian. For English, we use the 45K stories of Mostafazadeh et al. (2016) as the training set (see Table 4) and perform zero-shot cross-lingual transfer in both language directions (see Appendix for hyper-parameters).

For automatic evaluation we use ROUGE-L (Lin, 2004), BLEU-4 (Papineni et al., 2002), METEOR (Lavie and Agarwal, 2007), and BERTScore (Zhang et al., 2020). For Indonesian, we also conducted manual evaluation using 4 models × 50 randomly-sampled test instances, including gold sentences and predicted sentences, trained on the EN, ID, and EN+ID datasets. We asked two native speakers to read the premise and then examine whether the fifth sentence is coherent Indonesian text, does not contain repetition, follows commonsense, contains natural or unnatural code-switching

---

[6]We use the Huggingface Pytorch framework for fine-tuning (Wolf et al., 2019).

[7]https://translate.google.com/; accessed on April 2021.

| Context | $n$-gram | fastText | LSTM | MBERT | INDOBERT |
|---|---|---|---|---|---|
| None | — | — | $68.4 \pm 1.5$ | $75.7 \pm 0.9$ | $76.1 \pm 3.4$ |
| $s_4$ | 40.2 | 58.9 | $68.8 \pm 1.9$ | $77.1 \pm 1.4$ | $78.1 \pm 0.3$ |
| $s_3 \rightarrow s_4$ | 49.5 | 62.3 | $69.5 \pm 0.5$ | $77.3 \pm 1.5$ | $76.0 \pm 7.8$ |
| $s_2 \rightarrow s_4$ | **52.9** | 62.5 | $68.6 \pm 0.9$ | $77.8 \pm 0.9$ | $75.4 \pm 0.9$ |
| $s_1 \rightarrow s_4$ | 52.8 | **62.6** | $\mathbf{70.0 \pm 2.1}$ | $\mathbf{78.2 \pm 1.4}$ | $\mathbf{81.0 \pm 2.1}$ |

Table 5: Test classification accuracy (%) based on different contexts ($s_i$ indicates $i$-th sentence). Human accuracy is 99 (from 100 samples).

| Train | Test (EN) | Test (ID) |
|---|---|---|
| EN | $81.9 \pm 0.5$ | $71.3 \pm 2.3$ |
| ID | $68.1 \pm 1.9$ | $\mathbf{78.2 \pm 1.4}$ |
| EN+ID | $81.7 \pm 1.0$ | $76.8 \pm 1.1$ |
| EN$'$ | $69.2 \pm 1.5$ | $75.6 \pm 0.6$ |
| ID$'$ | $78.0 \pm 0.9$ | $69.6 \pm 0.4$ |
| EN+EN$'$ | $\mathbf{82.9 \pm 0.3}$ | $75.7 \pm 1.5$ |
| ID+ID$'$ | $78.6 \pm 0.6$ | $76.2 \pm 0.6$ |

Table 6: Test classification accuracy for English (EN) and Indonesian (ID) using MBERT. EN$'$ and ID$'$ indicate English and Indonesian translations, respectively, from Google Translate.

| Train | Test (EN) | | | | Test (ID) | | | |
|---|---|---|---|---|---|---|---|---|
| | R-L | B | M | BS | R-L | B | M | BS |
| EN | **20.4** | **6.9** | **9.2** | **75.2** | **19.2** | **6.6** | **8.2** | 73.8 |
| ID | 8.5 | 4.5 | 4.0 | 70.3 | 17.6 | 6.2 | 7.6 | 74.4 |
| EN+ID | 13.6 | 5.2 | 6.3 | 72.4 | 18.6 | 6.4 | 8.0 | **74.7** |

Table 7: Fifth-sentence generation using MBART over the test set (R-L, B, M, and BS indicate ROUGE-L, BLEU-4, METEOR, and BERTScore, respectively).

(in the case there is code-switching), and the overall story has good narrative flow.[8]

## 4 Results and Analysis

**Classification.** In Table 5, we find that a 1-sentence premise ($s_4$) is inadequate to comprehend the narrative of the story. We also observe that the $n$-gram method performs at near-random (52.9%), while fastText also struggles at 62.6% accuracy. The hierarchical BiLSTM and MBERT perform substantially better, at 70% and 78.2%, respectively.

Overall, the best performance is achieved by IN-DOBERT when using all sentences ($s_1 \rightarrow s_4$) as context, outperforming MBERT with 81% accu-

[8]Each worker was paid Rp 250,000.

| Train | A↑ | B↑ | C↑ | D↑ |
|---|---|---|---|---|
| Gold | 94 | 99 | 99 | 81 |
| EN | 72 | **66** | 58 | **31** |
| ID | **92** | 52 | 90 | 25 |
| EN+ID | **92** | 47 | **97** | **31** |

Table 8: Manual evaluation of the generation task for 50 randomly Indonesian samples, in terms of whether the fifth-sentence: **A**: does not contain repetition; **B**: follows commonsense; **C**: is fluent Indonesian; **D**: has good narrative flow. The presented scores are aggregated across two annotators (in %). The Kappa scores for each category range between 0.4–0.8 (see Appendix).

racy. Compared to the English Story Cloze Test, our corpus is arguably harder, as Li et al. (2019) reported BERT accuracies of 78% and 88.1% in the English corpus when using None and $s_1 \rightarrow s_4$ as the premise. We acknowledge that there is a spurious correlation of sentence-5 candidates with the commonsense labels, indicated by INDOBERT accuracy of 76.1% when having context of None. This phenomenon is worse in the English dataset (Mostafazadeh et al., 2016) where the BERT accuracy of using context of None is 88.1% (Li et al., 2019).

In Table 6, we use MBERT to examine commonsense reasoning crosslingually between English (EN) and Indonesian (ID). To simplify, we use L1→L2 to denote training in language L1 and testing in L2. First, we observe that combining EN and ID training worsens commonsense reasoning in both English and Indonesian. Applying zero-shot learning (i.e. EN→ID and ID→EN) achieves mixed results, and ID→EN has worse cross-lingual transfer than EN→ID in terms of performance gap over monolingual training. We argue this is because: (1) English is the dominant language in MBERT training, and (2) our ID corpus contains

contexts that are less universal (e.g. *nasi padang*[9] vs. *hamburger*).

To further observe whether the transferability is affected by factors beyond language, we translate the training data with Google Translate. In Table 6, EN′ denotes the English translation of the Indonesian training set, and ID′ vice versa. Surprisingly, we found that ID′ →ID has worse performance than EN→ID, while EN′ →EN improves slightly over ID→EN. This suggests that translating the training set to the test language is ineffective, and actually hurts performance for the ID test set. To further explore this effect, we asked two expert workers to evaluate 100 random sentences in the Google Translate output for EN–ID and ID–EN, and found quality in both translation directions to be high, with very little difference in terms of adequacy and fluency (4.5–4.6 out of 5).[10]

**Generation.** In Table 7, we observe that training using EN achieves the best performance across the automatic metrics on both the EN and ID test sets, with the one exception of BERTScore for EN+ID→ID.[11] However, in the manual evaluation of Indonesian (Table 8), we observe a different trend, in that training using the EN data tends to generate repetitive fifth sentences. Based on the manual evaluation, the best results are using ID and EN+ID as the training data, where the models do not suffer from repetition, generate fluent Indonesian, with similar acceptability in terms of commonsense reasoning.

Although zero-shot cross-lingual transfer of EN→ID suffers from repetition, we notice that MBART is capable of generating plausibly code-mixed sentences made up of Indonesian and English (Gardner-Chloros et al., 2009). Based on our manual evaluation on the same 50 Indonesian test set, we found that 41% of generated fifth sentences contain code-mixing, of which 75% are naturalistic (see Table 9 for examples).

## 5 Conclusion

In this paper, we introduced the first Indonesian story cloze dataset, and performed preliminary analysis in classification and generation settings in two scenarios: monolingual training and zero-shot cross-lingual transfer between Indonesian and

---

| **Natural code-mixing sentence** |
|---|
| *Now Armend memiliki printer di rumahnya* (Now Armend has a printer in his house) |
| *The only time Livia keluar kamar, adalah ketika ia sedang tidur* The only time Livia left the room is when she sleeps |

| **Unnatural code-mixing sentence** |
|---|
| *He Hendrik ditangkap oleh Polda* (He Hendrik is arrested by the local police) |
| *Shearing her teeth ketika diminta untuk menyanyi paling keras!* (Shearing her teeth when she is asked to sing loudly!) |

Table 9: Example of code-mixing sentence, generated by MBART when trained on the EN dataset. Red font denotes English words.

English. From both experiments, we found that the cross-lingual transfer of commonsense from English to Indonesian does not perform well, motivating the construction of commonsense reasoning resources in different languages.

## 6 Ethical Considerations

We paid our expert workers fairly, based on the monthly minimum wage in Indonesia. All workers were made aware that the submitted stories would be distributed, and used for research purposes. No sensitive information about the workers will be released.

## Acknowledgments

We are grateful to the anonymous reviewers for their helpful feedback and suggestions. The first author is supported by the Australia Awards Scholarship (AAS), funded by the Department of Foreign Affairs and Trade (DFAT), Australia.

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

## A  Training Configurations

### A.1  Classification

For LSTM, we set the maximum token for each sentence to be 30, and train the model for 100 epochs with early stopping (patience = 20), a batch size of 20, Adam optimizer, and a learning rate of 0.01. For pretrained-language model, we set the maximum token to be 450 and 50 for the premise and ending sentence, respectively, and train the model for 20 epochs with early stopping (patience = 5), a batch size of 40, Adam optimizer, an initial learning rate of 5e-5, and warm-up of 10% of the total steps.

### A.2  Generation

To train the sentence-5 generation task, we set the maximum length of tokens to be 200 and 50 for the input and target text, respectively. We train the models on 4×V100 32GB GPUs for 60 epochs with an initial learning rate of 1e-4 (Adam optimizer). We use a total batch size of 320 (20 x 4 GPUs x gradient accumulation of 4), a warmup of 10% of total steps, and save checkpoints for every 500 steps. We also compute ROUGE scores (R1) to pick the best checkpoint based on the development set. For calculating BERTScore we use `bert-base-multilingual-cased` based on layer suggested by Zhang et al. (2020).

## B  Additional Data Statistics

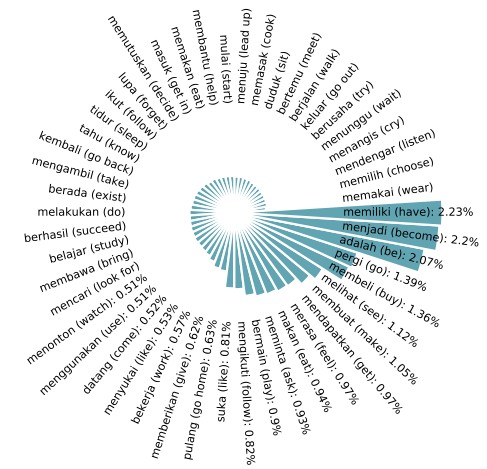

Figure 2: Distribution of top-50 verbs in our corpus.

## C  Analysis on Classification Task: FP and TP Samples

We further analyze false positive (FP) and true positive (TP) of INDOBERT by considering 1) whether the story contains temporal and causal relations; and 2) the number of premise sentences that are minimally required to entail the right ending.[12] We randomly selected 50 samples from each FP and TP sets, and found that 60% of FP samples have temporal relations while TP has lower percentage (56%). On the other hand, causal relations tends to be correctly predicted, with proportion 88% and 94% for FP and TP, respectively. Lastly,

---

[12]Sentence can be in any position.

we found that FP samples have a higher average of minimally-required premise: 2.8 (out of 4), while TP samples are only 2.1.

## D  Human Evaluations

| Aspect | Kappa Score |
|--------|-------------|
| A | 0.59 |
| B | 0.49 |
| C | 0.75 |
| D | 0.40 |
| E | 0.80 |
| F | 0.59 |

Table 10: **Generation task**: Kappa scores (inter-annotator agreement) of manual evaluation for 4 models × 50 randomly sampled Indonesian test. We evaluate whether the fifth-sentence: **A**: does not contain repetition; **B**: follows commonsense; **C**: is a fluent Indonesian; **D**: has a good flow; **E**: has natural English code-switching; and **F**: has unnatural English code-switching.

| Aspect | EN–ID | | ID–EN | |
|--------|----------|---------|----------|---------|
| | Adequacy | Fluency | Adequacy | Fluency |
| Pearson | 0.55 | 0.56 | 0.39 | 0.37 |
| Score | 4.47 | 4.57 | 4.60 | 4.58 |

Table 11: **Classification task**: We randomly sample 100 sentences (of stories) and use Google Translate to obtain the translation. We ask two expert workers to evaluate adequacy and fluency of EN–ID and ID–EN translation (Koehn and Monz, 2006). Scores reflect the average of two annotations, ranging between 1–5.

## E  Interview Questions

Buatlah sebuah cerita pendek dengan 5 kalimat!

Cerita pendek yang kami maksud terdiri dari 4 kalimat dan 2 kalimat penutup. Satu kalimat penutup merupakan kalimat yang sesuai dengan logika manusia berdasarkan 4 kalimat premise (sesuai dengan commonsense), sedangan 1 kalimat penutup lainnya merupakan kalimat yang tidak sesuai dengan logika manusia (commonsense).

==== Contoh STORY-1 ====

1. Nenek sangat suka menonton sinetron
2. Tiap sore setelah sholat isya beliau duduk di depan layar televisi selama 3 jam
3. Sesekali beliau bergumam karena kesal melihat pemeran antagonis yang tingkahnya sering menjahati pemeran utama
4. Tak jarang nenek juga ditemani kakek ketika menonton sinetron
Correct ending (5): Bagi nenek sinetron menjadi sarana hiburannya di malam hari
Incorrect ending (5): Nenek sangat ingin menjadi salah satu pemeran sinetron dan akan syuting esok hari

==== Contoh STORY-2 ====

1. Pak Miskin punya 3 orang anak
2. Sinta anak pertama kelas 6 SD
3. Anak kedua bernama Heru berusia 4 tahun
4. Anak yang paling kecil bernama Cahyono
Correct ending (5): Ia masih berusia 10 bulan
Incorrect ending (5): Cahyono duduk di kelas 3 SD

Make a short story with 5 sentences!

The short story consists of 4 sentences and 2 ending sentences. One ending sentence is a sentence that is in accordance with human logic based on 4 premise sentences (follows the commonsense), while the other one is a sentence that is not in accordance with human logic (do not follow the commonsense).

==== Example-1 ====

1. Grandma really likes watching soap operas.
2. Every evening after evening prayer she sits in front of the television for 3 hours.
3. Sometimes she muttered because she was annoyed to see the antagonist.
4. Often, she is accompanied by her husband when watching soap operas
Correct ending (5): For my grandmother, soap operas are a good entertainment at night
Incorrect ending (5): Grandma really wants to be a soap opera actor and will shoot tomorrow

==== Example-2 ====

1. Pak Miskin has 3 children
2. Sinta, the first child is in grade 6.
3. The second child named Heru is 4 years old
4. The youngest child is Cahyono
Correct ending (5): He is still 10 months old
Incorrect ending (5): Cahyono is in grade 3.

Figure 3: Interview question that is used in the hiring of story writers. The second row is the English translation (for illustration).

## F  Examples of Sentence-5 Generation

**Premise:**
*Sudah lima belas tahun Jerry tidak berkunjung ke SD tempatnya menuntut ilmu.*
*Saat ia akan menikah, ia mengunjungi sekolahnya untuk memberikan undangan ke guru-gurunya.*
*Saat bertemu mereka, ia merasa sangat terharu.*
*Guru-guru yang mengajarnya saat SD, kini tidak lagi semuda dulu.*

**Gold:**
*Meski begitu, mereka masih ingat dengan Jerry dan kenakalannya semasa sekolah*

**EN model:**
*Jerry merasa kehilangan sekolah tempatnya menuntut ilmu*

**ID model:**
*Jerry senang sekali dengan keberadaan guru-gurunya*

**EN+ID model:**
*Jerry sangat bangga dengan tempatnya belajar ilmu*

---

**Premise:**
It has been fifteen years that Jerry has not visited his elementary school.
Today he is visiting his school to invite his teachers to his wedding.
He feels so happy meeting with his former teachers.
Those teachers are no longer as young as fifteen years ago.

**Gold:**
Even so, they still remember Jerry.

**EN model:**
Jerry feels that he has lost his school.

**ID model:**
Jerry is very happy with his teachers.

**EN+ID model:**
Jerry is very proud of his primary school.

Figure 4: Example of sentence-5 generation output using MBART model. The second row is the English translation (for illustration).