# OpenReview forum: "Cloze Evaluation for Deeper Understanding of Commonsense Stories in Indonesian"
_aclweb.org/ACL/2022/Workshop/CSRR — ACL 2022 Workshop CSRR_

### Official Review · Reviewer_AtGC · 2022-03-22
**Story Cloze Task using Indonesian short stories**

**Rating:** 7
**Confidence:** 4

**Review:**

In this paper, the authors follow the Story Cloze Test framework of Mostafazadeh et al. (2016) in evaluating story understanding in Indonesian, by constructing a four-sentence story with one correct ending and one incorrect ending. To investigate commonsense knowledge acquisition in language models, authors experimented with: (1) a classification task to predict the correct ending; and (2) a generation task to complete the story with a single sentence. They investigate these tasks in two settings: (i) monolingual training and (ii) zero-shot cross-lingual transfer between Indonesian and English.

1) The authors put a considerable amount of effort into ensuring the quality of the data. Even though it's not huge it's still an important resource towards multilingual commonsense knowledge.
2) Authors consider both generative and discriminative settings and use extensive baselines and particular both monolingual and cross lingual methods
3) For classification the ablation with different contexts is really neat and gives the reader an idea of the difficulty of the dataset. 81% is still a lot lower than human performance. However, the high performance without showing context shows the dataset might not be free from annotation artifacts
4) For the generation I appreciate authors showing both human and automatic evaluation. The human eval results for presence of commonsense and good narrative flow is slightly concerning
5) The authors should mention the agreement of human evaluations

---

### Official Review · Reviewer_1PxG · 2022-03-22
**Contribution of an interesting story cloze dataset in Indonesian; comprehensive experiments against multiple baselines.**

**Rating:** 8
**Confidence:** 4

**Review:**

This paper creates a story cloze dataset in the form of ROCStories in Indonesian. The authors present analysis on the dataset indicating the presence of cultural entities/names/etc. that highlight the importance of collecting datasets in the desired language directly rather than simply translating the English dataset.

The authors present results from a number of baseline models---n-gram similarity, fastText embedding similarity, mBERT, and IndoBERT--- on the last-sentence-selection classification task, demonstrating that the naive similarity methods are not significantly better than random, while IndoBERT achieves 81% accuracy, indicating the task is not solved. They perform analysis to show that the dataset does not have the artifacts that the original ROCStories dataset has-- experiments show that the full context (first 4 sentences) is generally needed to achieve the best performance on the classification task. However, I'd ask the authors to acknowledge in the text that the models receiving no context are still far better than the 50% baseline, indicating some form of spurious correlations in the dataset.

On the last-sentence generation task, the authors show that training on both the English + Indonesian data leads to good performance on the Indonesian test set using mBART, according to manual evaluation. They also test zero-shot cross-lingual transfer on the classification task by training and testing on different variants of their dataset (Indonesian), ROCStories (English), and machine translations of each, finding that both the English and Indonesian test sets don't seem to benefit from training on the other language, and including machine translations has mixed effect.

The data is an interesting and novel contribution, and the inclusion of detailed experiments makes promising progress on the task with high-quality analysis. My only concern is that last-sentence prediction and generation from 5-sentence stories are very simplified versions of creative storytelling, and many works in the research area (albeit in English) now focus on harder tasks such as generating entire stories from a single prompt or from scratch on more complex datasets. However I think this resource could be used for such goals and is thus useful to the community as a starting point.

Possible missing citation: Abductive Commonsense Reasoning-- a variant of the story cloze task (https://openreview.net/pdf?id=Byg1v1HKDB)

---

### Official Review · Reviewer_5PLE · 2022-03-24
**Applied, Practical, Introduced new dataset, Multiple ablation studies are reported**

**Rating:** 7
**Confidence:** 3

**Review:**

This paper investigates commonsense knowledge acquisition in language models via classification and generation tasks.

[Strong]
Authors introduced  the first Story Cloze Test in Indonesian. Highly practical applied work with a lot of useful practical details. Detailed ablation results are presented.

[Weak]
Not novel

---

### Decision · Program_Chairs · 2022-03-28

Accept